# Machine Learning Estimation of Heterogeneous Treatment Effects with Instruments

**Vasilis Syrgkanis**
Microsoft Research
vasy@microsoft.com

**Victor Lei**
TripAdvisor
vlei@tripadvisor.com

**Miruna Oprescu**
Microsoft Research
moprescu@microsoft.com

**Maggie Hei**
Microsoft Research
Maggie.Hei@microsoft.com

**Keith Battocchi**
Microsoft Research
kebatt@microsoft.com

**Greg Lewis**
Microsoft Research
glewis@microsoft.com

## Abstract

We consider the estimation of heterogeneous treatment effects with arbitrary machine learning methods in the presence of unobserved confounders with the aid of a valid instrument. Such settings arise in A/B tests with an intent-to-treat structure, where the experimenter randomizes over which user will receive a recommendation to take an action, and we are interested in the effect of the downstream action. We develop a statistical learning approach to the estimation of heterogeneous effects, reducing the problem to the minimization of an appropriate loss function that depends on a set of auxiliary models (each corresponding to a separate prediction task). The reduction enables the use of all recent algorithmic advances (e.g. neural nets, forests). We show that the estimated effect model is robust to estimation errors in the auxiliary models, by showing that the loss satisfies a Neyman orthogonality criterion. Our approach can be used to estimate projections of the true effect model on simpler hypothesis spaces. When these spaces are parametric, then the parameter estimates are asymptotically normal, which enables construction of confidence sets. We applied our method to estimate the effect of membership on downstream webpage engagement on TripAdvisor, using as an instrument an intent-to-treat A/B test among 4 million TripAdvisor users, where some users received an easier membership sign-up process. We also validate our method on synthetic data and on public datasets for the effects of schooling on income.[1]

## 1 Introduction

A/B testing is the gold standard of causal inference. But even when A/B testing is feasible, estimating the effect of a treatment on an outcome might not be a straightforward task. One major difficulty is non-compliance: even if we randomize what treatment to recommend to a subject, the subject might not comply with the recommendation due to unobserved factors and follow the alternate action. The impact that unobserved factors might have on the measured outcome is a source of endogeneity and can lead to biased estimates of the effect. This problem arises in large scale data problems in the digital economy; when optimizing a digital service, we might often want to estimate the effect of some action taken by our users on downstream metrics. However, the service cannot force users to comply, but can only find means of incentivizing or recommending the action. The unobserved factors of compliance can lead to biased estimates if we consider the takers and not takers as exogenously

assigned and employ machine learning approaches to estimate the potentially heterogeneous effect of the action on the downstream metric.

The problem can be solved by using the technique of instrumental variable (IV) regression: as long as the recommendation increases the probability of taking the treatment, then we know that there is at least some fraction of users that were assigned the treatment "exogeneously". IV regression parses out this population of "exogenously treated" users and estimates an effect based solely on them.

Most classical IV approaches estimate a constant average treatment effect. However, to make personalized policy decisions (an emerging trend in most digital services) one might want to estimate a heterogeneous effect based on observable characteristics of the user. The latter is a daunting task, as we seek to estimate a function of observable characteristics as opposed to a single number. Hence, statistical power is at stake. Even estimating an ATE is non-trivial when effect and compliance are correlated through observables. The emergence of large data-sets in the digital economy alleviates this concern; with A/B tests running on millions of users it is possible to estimate complex heterogeneous effect models, even if compliance levels are relatively weak. Moreover, as we control for more and more observable features of the user, we also reduce the risk that correlation between effect and compliance is stemming from unobserved factors.

This leads to the question this work seeks to answer: how can we blend the power of modern machine learning approaches (e.g. random forests, gradient boosting, penalized regressions, neural networks) with instrumental variable methods, so as to estimate complex heterogeneous effect rules. Recent work at the intersection of machine learning and econometrics has proposed powerful methods for estimating the effect of a treatment on an outcome, while using machine learning methods for learning nuisance models that help de-bias the final effect rule. However, the majority of the work has either focused on 1) estimating average treatment effects or low dimensional parametric effect models (e.g. the double machine learning approach of [11]), 2) developing new algorithms for estimating fully non-parametric models of the effect (e.g. the IV forest method of [4], the DeepIV method of [15]), 3) assuming that the treatment is exogenous once we condition on the observable features and reducing the problem to an appropriate square loss minimization framework (see e.g. [26, 21]).

Nevertheless, a general reduction of IV based machine learning estimation of heterogeneous effects to a more standard statistical learning problem that can incorporate existing algorithms in a black-box manner has not been formulated in prior work. In fact, the recent work of [26], which develops a statistical learning based approach in the setting with no unboserved confounders, leaves as a major open question the development of an analogue statistical learning approach for our setting, with unobserved confounders and access to valid instruments. Such a reduction can help us leverage the recent algorithmic advances in statistical learning theory so as to work with large data-sets. Our work proposes the reduction of heterogeneous effects estimation via instruments to a square loss minimization problem over a hypothesis space. This enables us to learn not only the true heterogeneous effect model, but also the projections of the true model in simpler hypothesis spaces for interpretability. Moreover, our work leverages recent advances in statistical learning with nuisance functions [12, 13], to show that the mean squared error (MSE) of the learned model is robust to the estimation error of auxiliary models that need to be estimated (as is standard in IV regression). Thus we achieve MSE rates where the leading term depends only on the sample complexity of the hypothesis space of the heterogeneous effect model.

Some advantages of reducing our problem to a set of standard regression problems include being able to use existing algorithms and implementations, as well as recent advances of interpretability in machine learning. For instance, in our application we deploy the SHAP framework [23, 22] to interpret random forest based models of the heterogeneous effect. Furthermore, when the hypothesis space is low dimensional and parametric then our approach falls in the setting studied by prior work of [11] and, hence, not only MSE rates but also confidence interval construction is relatively straightforward. This enables hypothesis testing on parametric projections of the true effect model.

We apply our approach to an intent-to-treat A/B test among 4 million users on a major travel webpage so as to estimate the effect of membership on downstream engagement. We identify sources of heterogeneity that have policy implications on which users the platform should engage more and potentially how to re-design the recommendation to target users with large effects. We validate the findings on a different cohort in a separate experiment among 10 million users on the same platform. Even though the new experiment was deployed on a much broader and different cohort, we identify common leading factors of effect heterogeneity, hence confirming our findings. As a robustness check

we create semi-synthetic data with similar features and marginal distributions of variables as the real data, but where we know the ground truth. We find that our method performs well both in terms of MSE, identifying the relevant factors and coverage of the confidence intervals.

Finally, we apply our method to a more traditional IV application: estimating the effect of schooling on wages. We use a well studied public data set and observe that our approach automatically identifies sources of heterogeneity that were previously uncovered using more structural approaches. We also validate our method in this application on semi-synthetic data that emulate the true data.

## 2 Estimation of Heterogeneous Treatment Effects with Instruments

We consider estimation of heterogeneous treatment effects with respect to a set of features $X$, of an endogenous treatment $T$ on an outcome $Y$ with an instrument $Z$. For simplicity of exposition, we will restrict attention to the case where $Y, Z$ and $T$ are scalar variables, but several of our results extend to the case of multi-dimensional treatments and instruments. $Z$ is an instrumental variable if it has an effect on the treatment but does not have a direct effect on the outcome other than through the treatment. More formally, we assume the following moment condition:

$$\mathbb{E}[Y - \theta_0(X)T - f_0(X) \mid Z, X] = 0 \tag{1}$$

Equivalently we assume that: $Y = \theta_0(X)\,T + f_0(X) + e$, with $\mathbb{E}[e \mid Z, X] = 0$. We allow for the presence of confounders, i.e. $e$ could be correlated with $T$ via some unobserved common factor $\nu$. However, our exclusion restriction on the instrument implies that the residual is mean zero conditional on the instrument. Together with the fact that the instrument also has an effect on the treatment at any value of the feature $X$, i.e.: $\mathrm{Var}(\mathbb{E}[T \mid Z, X] \mid X) \geq \lambda$, allows us to identify the heterogeneous effect function $\theta_0(X)$. We focus on the case where the effect is linear in the treatment $T$, which is wlog in the binary treatment setting, which is our main application, and since our goal is to focus on the non-linearity wrt $X$ (this greatly simplifies our problem, see [9, 10, 25, 16]).[2]

Given $n$ i.i.d. samples from the data generating process, our goal is to estimate a model $\hat{\theta}(X)$ that achieves small expected mean-squared-error, i.e.: $\mathbb{E}[\|\hat{\theta} - \theta_0\|_2] := \mathbb{E}[(\hat{\theta}(X) - \theta_0(X))^2] \leq R_n$. Since the true $\theta_0$ function can be very complex and difficult to estimate in finite samples, we are also interested in estimating projections of the true $\theta_0$ on simpler hypothesis spaces $\Theta_\pi$. Projections are also useful for interpretability: one might want to understand what is the best linear projection of $\theta_0(X)$ on $X$, i.e. $\alpha_0 = \arg\min_\alpha \mathbb{E}[(\langle \alpha, X \rangle - \theta_0(X))^2]$. In this case we will denote with $\theta_*$ the projection of $\theta_0$ on $\Theta_\pi$, i.e. $\theta_* = \arg\min_{\theta \in \Theta_\pi} \mathbb{E}[(\theta(X) - \theta_0(X))^2]$ and our goal would be to achieve small mean squared error with respect to $\theta_*$. When $\theta_*$ is a low dimensional parametric class (e.g. a linear function on a low-dimensional feature space or a constant function), we are also interested in performing inference; i.e. constructing confidence intervals that asymptotically contain the correct parameter with probability equal to some target confidence level.

**Warm-Up: Estimating the Average Treatment Effect (ATE)** For estimation of the average treatment effect (ATE), *assuming that either there is no effect heterogeneity with respect to $X$ or there is no heterogeneous compliance with respect to $X$*, [11] propose a method for estimating the ATE that solves the empirical analogue of the following moment equation:

$$\mathbb{E}[(Y - \mathbb{E}[Y \mid X] - \theta(T - \mathbb{E}[T \mid X]))\,(Z - \mathbb{E}[Z \mid X])] = 0 \tag{2}$$

This moment function is orthogonal to all the functions $q_0(X) = \mathbb{E}[Y \mid X]$, $p_0(X) = \mathbb{E}[T \mid X]$ and $r_0(X) = \mathbb{E}[Z \mid X]$ that also need to be estimated from data. This moment avoids the estimation of the expected $T$ conditional on $Z, X$ and satisfies an orthogonality condition that enables robustness of the estimate $\hat{\theta} = \frac{\mathbb{E}_n[(Y - \hat{q}(X))\,(Z - \hat{r}(X))]}{\mathbb{E}_n[(T - \hat{p}(X))\,(Z - \hat{r}(X))]}$, to errors in the nuisance estimates $\hat{q}, \hat{r}$ and $\hat{p}$. The estimate is asymptotically normal with variance equal to the variance of the method if the estimates were the

correct ones, assuming that the mean squared error of these estimates decays at least at a rate of $n^{-1/4}$ (see [11] for more details). This result requires that the nuisance estimates are fitted in a *cross-fitting* manner, i.e. we use half of the data to fit a model for each of these functions and then predict the values of the model on the other half of the samples. We refer to this algorithm as DMLATEIV.[3]

**Inconsistency under Effect and Compliance Heterogeneity** The above estimate $\hat{\theta}$ is a consistent estimate of the average treatment effect as long as there is either no effect heterogeneity with respect to $X$ or there is no heterogeneous compliance (i.e. the effect of the instrument on the treatment) with respect to $X$. Otherwise it is inconsistent. The reason is that, if we let $\tilde{T} = T - p_0(X)$ and $\tilde{Z} = Z - r_0(X)$, then the population quantity: $\beta_0(X) = \mathbb{E}[\tilde{T}\tilde{Z} \mid X]$ is a function of $X$. If we also have effect heterogeneity, then we are solving for a constant $\hat{\theta}$ that in the limit satisfies: $\mathbb{E}[(\tilde{Y} - \hat{\theta}\tilde{T})\tilde{Z}] = 0$, where $\tilde{Y} = Y - q_0(X)$. On the other hand the true heterogeneous model satisfies the equation: $\mathbb{E}[(\tilde{Y} - \theta_0(X)\tilde{T})\tilde{Z}] = 0$. In the limit, the two quantities are related via the equation: $\hat{\theta}\,\mathbb{E}[\tilde{T}\tilde{Z}] = \mathbb{E}[\theta_0(X)\tilde{T}\tilde{Z}]$. Then the constant effect that we estimate converges to the quantity: $\hat{\theta} = \frac{\mathbb{E}[\theta_0(X)\beta_0(X)]}{\mathbb{E}[\beta_0(X)]}$. If $\theta_0(X)$ is not independent with $\beta_0(X)$, then $\hat{\theta}$ is a re-weighted version of the true average treatment effect $\mathbb{E}[\theta(X)]$, re-weighted by the heterogeneous compliance. To account for this heterogeneous compliance we need to change our moment equation so as to re-weight based on $\beta_0(X)$, which is unknown and also needs to be estimated from data. Given that this function could be arbitrarily complex, we want our final estimate to be robust to estimation errors of $\beta_0(X)$. We can achieve this by considering a doubly robust approach to estimating $\hat{\theta}$. Suppose that we had some other method of computing an estimate of the heterogeneous treatment effect $\theta_0(X)$, then we can combine both estimates to get a more robust method for the ATE, e.g.:

$$\hat{\theta}_{DR} = \mathbb{E}\left[\hat{\theta}(X) + \frac{(\tilde{Y} - \hat{\theta}(X)\tilde{T})\tilde{Z}}{\hat{\beta}(X)}\right] \qquad (3)$$

This approach has been analyzed in [27] in the case of constant treatment effects and an analogue of this average effect was also used by [5] in a policy learning problem as opposed to an estimation problem. In particular, the quantity $\tilde{Z}/\beta(X)$ is known as the compliance score [1, 3]. Our methodological contribution in the next two sections is two-fold: i) first we propose a model-based stable approach for estimating a preliminary estimate $\hat{\theta}(X)$, which does not necessarily require that $\beta(X) > 0$ everywhere (an assumption that is implicit in the latter method), ii) second we show that this doubly robust quantity can be used as a regression target and minimizing the square loss with respect to this target, corresponds to an orthogonal loss, as defined in [12, 13].

## 2.1 Preliminary Estimate of Conditional Average Treatment Effect (CATE)

Let $h_0(Z, X) = \mathbb{E}[T \mid Z, X]$ and $p_0, q_0$ as in the previous section. Then observe that we can re-write the moment condition as: $\mathbb{E}[Y - \theta_0(X)\,h_0(Z, X) - f_0(X) \mid Z, X] = 0$. Moreover, observe that the functions $p_0, q_0$ and $f_0$ are related via: $q_0(X) = \theta_0(X)\,p_0(X) + f_0(X)$. Thus we can further re-write the moment condition in terms of $q_0, p_0$ instead of $f_0$: $\mathbb{E}[Y - q_0(X) - \theta_0(X)\,(h_0(Z, X) - p_0(X)) \mid Z, X] = 0$. Moreover, we can identify $\theta(X)$ with the following subset of conditional moments, where the conditioning of $Z$ is removed: $\mathbb{E}[(Y - q_0(X) - \theta(X)\,(h_0(Z, X) - p_0(X)))\,(h_0(Z, X) - p_0(X)) \mid X] = 0$. Equivalently, $\theta(X)$ is a minimizer of the square loss:

$$L^1(\theta; q_0, h_0, p_0) := \mathbb{E}\left[(Y - q_0(X) - \theta(X)\,(h_0(Z, X) - p_0(X)))^2\right] \qquad (4)$$

since the derivative of this loss with respect to $\theta(X)$ is equal to the moment equation and, thus, the first order condition for the loss minimization problem is satisfied by the true model $\theta_0$. Moreover, if the loss function satisfies a functional analogue of strong convexity, then any minimizer of the loss

achieves small mean squared error with respect to $\theta_0$. This leads to the following approach:

---

**Algorithm 1:** HETEROGENEOUS EFFECTS: DMLIV Partially orthogonal, convex loss.

**1** On a half-sample $S_1$: regress i) $Y$ on $X$, ii) $T$ on $X, Z$, iii) $T$ on $X$, to learn estimates $\hat{q}$, $\hat{h}$ and $\hat{p}$ corr.;
**2** Minimize the empirical analogue of the square loss over some hypothesis space $\Theta$ on the other half-sample $S_2$:

$$\hat{\theta} = \arg\inf_{\theta \in \Theta} \frac{2}{n} \sum_{i \in S_2} (Y_i - \hat{q}(X_i) - \theta(X_i)(\hat{h}(Z_i, X_i) - \hat{p}(X_i)))^2 := L_n^1(\theta; \hat{q}, \hat{h}, \hat{p}) \qquad (5)$$

or any learning algorithm that achieves small generalization error w.r.t. loss $L^1(\theta; \hat{q}, \hat{h}, \hat{p})$ over $\Theta$.

---

This method is an extension of the classical two-stage-least-squares (2SLS) approach [2] to allow for arbitrary machine learning models; ignoring the residualization part (i.e. if for instance $q(X) = p(X) = 0$), then it boils down to: 1) predict the mean treatment from the instrument and $X$ with an arbitrary regression/classification method, 2) predict the outcome from the predicted treatment multiplied by the heterogeneous effect model $\theta(X)$. Residualization helps us remove the dependence of the mean squared error on the complexity of the baseline function $f_0(X)$. We achieve this by showing that this loss is orthogonal with respect to $p, q$ (see [13] for the definition of an orthogonal loss). However, orthogonality does not hold with respect to $h$. This finding is reasonable since we are using $h(Z, X)$ as our regressor. Hence, any error in the measurement of the regressor can directly propagate to an error in $\theta(X)$. This is the same reason why in classical IV regression one cannot ignore the variance from the first stage of 2SLS when calculating confidence intervals.

**Lemma 1.** *The loss function $L^1(\theta; q, h, p)$ is orthogonal to the nuisance functions $p, q$, but not $h$.*

**Strong convexity and overlap.** Note that both the empirical loss $L_n^1$ and the population loss $L^1$ are convex in the prediction, which typically implies computational tractability. Moreover, the second order directional derivative of the population loss in any functional direction $\theta(\cdot) - \theta_0(\cdot)$ is: $\mathbb{E}\left[(\hat{h}(Z, X) - \hat{p}(X))^2 (\theta(X) - \theta_0(X))^2\right]$ and let: $V(X) := \mathbb{E}\left[(\hat{h}(Z, X) - \hat{p}(X))^2 \mid X\right]$. To be able to achieve mean-squared-error rates based on our loss minimization, we need the population version $L^1$ of the loss function to satisfy a functional analogue of $\lambda$-strong convexity:

$$\forall \theta \in \Theta : \mathbb{E}[V(X) \cdot (\theta(X) - \theta_0(X))^2] \geq \lambda \,\mathbb{E}[(\theta(X) - \theta_0(X))^2] \qquad (6)$$

This setting falls under the "single-index" setup of [13]. Using arguments from Lemma 1 of [13], if:

$$\forall \theta \in \Theta : \mathbb{E}[V_0(X) \cdot (\theta(X) - \theta_0(X))^2] \geq \lambda_0 \,\mathbb{E}[(\theta(X) - \theta_0(X))^2] \qquad (7)$$

where $V_0(X) := \mathbb{E}\left[(h_0(Z, X) - p_0(X))^2 \mid X\right] = \text{Var}(\mathbb{E}[T \mid Z, X] \mid X)$, then $\lambda \geq \lambda_0 - O(\|h - h_0\|_4^2, \|p - p_0\|_4^2) = \lambda_0 - o(1)$. A sufficient condition is that $V_0(X) \geq \lambda_0$ for all $X$. This is a standard "overlap" condition that the instrument is exogenously varying at any $X$ and has a direct effect on the treatment at any $X$. DMLIV only requires an "average" overlap condition, tailored particularly to the hypothesis space $\Theta$, hence it could handle settings where the instrument is weak for some subset of the population. For instance, if $\Theta$ is a linear function class: $\Theta = \{\langle \theta, \phi(X) \rangle : \theta \in S \subseteq \mathbb{R}^d\}$, then for the oracle strong convexity to hold it suffices that: $\mathbb{E}[V(X)\phi(X)\phi(X)^T] \succeq \lambda I$. Lemma 1, combined with the above discussion and the results of [13] yields:[4]

**Corollary 2.** *Assume all random variables are bounded and consider any algorithm that achieves expected generalization error $R_n^2$ with respect to loss $L^1(\theta; \hat{q}, \hat{h}, \hat{p})$. Moreover, suppose that the nuisance estimates satisfy $\|\hat{q} - q_0\|_4, \|\hat{p} - p_0\|_4 = o(g_n)$ and $\|\hat{h} - h_0\|_4 = o(h_n)$. Then $\hat{\theta}$ returned by DMLIV satisfies: $\|\hat{\theta} - \theta_0\|_2^2 \leq O\left(\frac{R_n^2 + h_n^2 + g_n^4}{\lambda_0}\right)$. If empirical risk minimization is used in the final stage, then $R_n^2 = \delta_n^2 + h_n^2 + g_n^4$, where $\delta_n$ is the critical radius of the hypothesis space $\Theta$ as defined via the localized Rademacher complexity [20].*

**Computational considerations.** The empirical loss $L_n^1$ is not a standard square loss. However, we can re-write it as $\sum_i \gamma(X_i)^2 (\tilde{Y}_i/\gamma(X_i) - \theta(X_i))^2$. Thus the problem is equivalent to a standard square loss minimization with label $\tilde{Y}_i/\gamma(X_i)$ and sample weights $\gamma(X_i)^2$. Thus we can use any out-of-the-box machine learning method that accepts sample weights, such as stochastic gradient based regression methods and gradient boosted or random forests. Alternatively, if we assume a linear representation of the effect function $\theta(X) = \langle \theta, \phi(X) \rangle$, then the problem is equivalent to regressing $\tilde{Y}$ on the scaled features $\phi(X)\gamma(X)$, and again any method for fitting linear models can be invoked.

## 2.2 DRIV: Orthogonal Loss for IV Estimation of CATE and Projections

We now present the main estimation algorithm that combines the doubly robust approach presented for ATE estimation with the preliminary estimator of the CATE to obtain a fully orthogonal and strongly convex loss. This method achieves a second order effect from all nuisance estimation errors and enables oracle rates for the target effect class $\Theta$ and asymptotically valid inference for low dimensional target effect classes. In particular, given access to a first stage model of heterogeneous effects $\theta_{pre}$ (such as the one produced by DMLIV), we can estimate a more robust model of heterogeneous effects via minimizing a square loss that treats the doubly robust quantity used in Equation (3) as the label:

$$\min_{\theta \in \Theta_\pi} L^2(\theta; \theta_{pre}, \beta, p, q, r) := \mathbb{E}\left[\left(\theta_{pre}(X) + \frac{(\tilde{Y} - \theta_{pre}(X)\tilde{T})\tilde{Z}}{\beta(X)} - \theta(X)\right)^2\right] \qquad (8)$$

We allow for a model space $\Theta_\pi$ that is not necessarily equal to $\Theta$. The solution in Equation (3) is a special case of this minimization problem where the space $\Theta_\pi$ contains only constant functions. Our main result shows that this loss is orthogonal to all nuisance functions $\theta_{pre}, \hat{\beta}, \hat{q}, \hat{p}, \hat{r}$. Moreover, it is strongly convex in the prediction $\theta(X)$, since conditional on all the nuisance estimates it is a standard square loss. Moreover, we show that the loss is orthogonal irrespective of what the model space $\Theta_\pi$, even if $\Theta_\pi \neq \Theta$, as long as the preliminary estimate $\theta_{pre}$ is consistent with respect to the true CATE $\theta_0$ (i.e. fit a flexible preliminary CATE and use it to project to a simpler hypothesis space).

**Lemma 3.** *The loss $L^2$ is orthogonal with respect to the nuisance functions $\theta_{pre}$, $\beta$, $p$, $q$ and $r$.*

---
**Algorithm 2:** DRIV Orthogonal convex loss for CATE and projections of CATE

---
1 Estimate a preliminary estimate $\theta_{pre}$ of the CATE $\theta_0(X)$ using DMLIV on half-sample $S_1$;
2 Using half-sample $S_1$, regress i) $Y$ on $X$, ii) $T$ on $X$, iii) $Z$ on $X$ to learn estimates $\hat{q}, \hat{p}, \hat{r}$ correspondingly;
3 Regress $T \cdot Z$ on $X$ using $S_1$ to learn estimate $\hat{f}$ of function $f_0(X) = \mathbb{E}[T \cdot Z \mid X]$;
4 $\forall i \in S_2$, let $\tilde{Y}_i = Y_i - \hat{q}(X_i)$, $\tilde{T}_i = T_i - \hat{p}(X_i)$, $\tilde{Z}_i = Z_i - \hat{r}(X_i)$, $\hat{\beta}(X_i) = \hat{f}(X_i) - \hat{p}(X_i)\,\hat{r}(X_i)$;
5 Minimize empirical analogue of square loss $L^2$ over hypothesis space $\Theta_\pi$ on the other half-sample $S_2$, i.e.:

$$\hat{\theta}_{DR} = \arg\inf_{\theta \in \Theta_\pi} \frac{2}{n} \sum_{i \in S_2}\left(\theta_{pre}(X_i) + \frac{(\tilde{Y}_i - \hat{\theta}(X_i)\tilde{T}_i)\tilde{Z}_i}{\hat{\beta}(X_i)} - \theta(X_i)\right)^2 := L_n^2(\theta; \theta_{pre}, \hat{\beta}, \hat{p}, \hat{q}, \hat{r})$$

or any learning algorithm that has small generalization error w.r.t. loss $L^2(\theta; \theta_{pre}, \hat{\beta}, \hat{p}, \hat{q}, \hat{r})$ on $\Theta_\pi$.

---

If we use DMLIV for $\theta_{prel}$, even though DMLIV has a first order impact from the error of $h$, the second stage estimate has a second order impact, since it has a second order impact from the first stage CATE error. Lemma 3 together with the results of [13] and [12] implies the following corollary:

**Corollary 4.** *Assume all random variables are bounded and consider any algorithm that achieves expected generalization error $R_n^2$ with respect to loss $L^2(\theta; \theta_{pre}, \hat{\beta}, \hat{p}, \hat{q}, \hat{r})$. Moreover, suppose that each nuisance estimate $\hat{g} \in \{\theta_{pre}, \hat{\beta}, \hat{p}, \hat{q}, \hat{r}\}$, $\|\hat{g} - g_0\|_4 \leq g_n$. Then $\hat{\theta}$ returned by DRIV satisfies: $\|\hat{\theta} - \theta_*\|_2^2 \leq O\left(R_n^2 + g_n^4\right)$, where $\theta_* = \arg\min_{\theta \in \Theta_\pi} L^2(\theta; \theta_0, \beta_0, p_0, q_0, r_0)$. If empirical risk minimization is used in the final stage, then $R_n^2 = \delta_n^2 + g_n^4$, where $\delta_n$ is the critical radius of the hypothesis space $\Theta$ as defined via the localized Rademacher complexity [20]. If $\Theta$ is high-dimensional sparse linear, i.e. $\theta(X) = \langle \xi, \phi(X) \rangle$ with $\|\xi\|_0 \leq s$, $\phi(X) \in \mathbb{R}^p$ and $\mathbb{E}[\phi(X)\phi(X)^T] \geq \lambda_0 I$, then if an $\ell_1$-penalized square loss minimization is used in the final step of DRIV, it suffices that $\|\hat{g} - g_0\|_2 \leq g_n$ to get: $\|\hat{\xi} - \xi_*\|_2^2 \leq O\left(s^2 \frac{\log(p)/n + g_n^4}{\lambda_0}\right)$.*

**Interpretability through projections.** The fact that our loss function can be used with any target $\Theta_\pi$ allows us to perform inference on the projection of $\theta_0$ on a simple space $\Theta_\pi$ (e.g. decision trees, linear functions) for interpretability purposes. If we let $Y_i^{DR}$ the label in the final regression of DRIV, then observe that when the nuisance estimates take their true values then $\mathbb{E}[Y_i^{DR} \mid X] = \theta_0(X)$, since the second part of $Y_i^{DR}$ has mean zero. Hence: $L^2(\theta; \theta_0, \beta_0, p_0, q_0, r_0) = \text{Var}(\theta_{DR}(X)) + \mathbb{E}[(\theta_0(X) - \theta(X))^2]$. The first part is independent of $\theta$ and hence minimizing the oracle $L^2$ is equivalent to minimizing $\mathbb{E}[(\theta_0(X) - \theta(X))^2]$ over $\theta \in \Theta_\pi$, which is exactly the projection of $\theta_0$ on $\Theta_\pi$. One version of an interpretable model is estimating the CATE with respect to a subset $T$ of the variables, i.e.: $\theta(X_T) = \mathbb{E}[\theta_0(X) \mid X_T]$ (e.g. how treatment effect varies with a single feature). This boils down to setting $\Theta_\pi$ some space of functions of $X_T$.

If $T$ is a low dimensional set of features and $\Theta_\pi$ is a the space of linear functions of $X_T$, i.e. $\Theta_\pi = \{X \to \langle \theta_T, X_T \rangle : \theta_T \in \mathbb{R}^{|T|}\}$, then the first order condition of our loss is equal to the

moment condition $\mathbb{E}[(Y^{DR} - \langle\theta_T, X_T\rangle)X_T] = 0$. Then orthogonality of our loss implies that DRIV is equivalent to an orthogonal moment estimation method [11]. Thus using the results of [11] we get that the estimate $\hat{\theta}_T$ of DRIV is asymptotically normal with asymptotic variance equal to the hypothetical variance of $\theta_T$ as if the nuisance estimates had their true values. Hence, we can use out-of-the-box packages for calculating CIs of an OLS regression to get $p$-values on the coefficients.

## 3 Estimating Effects of Membership at TripAdvisor

We apply our methods to estimate the treatment effect of membership on the number of days a user visits TripAdvisor. The instrument used was a 14-day intent-to-treat A/B test run during 2018, where users in group A received a new, easier membership sign-up process, while the users in group B did not. The treatment is whether a user became a member or not. Becoming a member and logging into TripAdvisor gives users exclusive access to trip planning tools, special deals and price alerts, and personalized ideas and travel advice. Our data consists of 4,606,041 total users in a 50:50 A/B test. For each user, we have a 28-day pre-experiment summary about their browsing and purchasing activity on TripAdvisor (see Sec. B.2). The instrument significantly increased the rate of treatment, and is assumed to satisfy the exclusion restriction.

We applied two sets of nuisance estimation models with different complexity characteristics: LASSO regression and logistic regression with an L2 penalty (LM); and gradient boosting regression and classification (GB). The only exception was $E[Z|X]$, where we used a fixed estimate of $0.5$ since the instrument was a large randomized experiment. See Sec. B.1 for details.[5]

| Nuisance | Method | ATE Est [95% CI] | Nuisance | Method | ATE Est [95% CI] |
|---|---|---|---|---|---|
| LM | DMLATEIV | 0.117 [-0.051, 0.285] | GB | DMLATEIV | 0.127 [-0.031, 0.285] |
| LM | DRIV | 0.113 [-0.052, 0.279] | GB | DRIV | 0.125 [-0.061, 0.311] |

Table 1: ATE Estimates for 2018 Experiment at TripAdvisor

We estimate the ATE using DRIV projected onto a constant (Table 1). Using linear nuisance models results in very similar ATE estimates between DMLATEIV and DRIV. We compare the $X$ co-variate associations for both heterogeneity and compliance under DRIV to understand why. If there are co-variates with significant non-zero associations in both heterogeneity and compliance, this could lead to different estimates between DRIV and DMLATEIV (and vice versa). Replacing the CATE projection model with a linear regression, we obtain valid inferences for the co-variates associated with treatment effect heterogeneity (Figure 1). For compliance, we run a linear regression of the estimated quantity $\beta(X)$ on $X$, to assess its association with each of the features (see Sec. B.1 for details). Comparing treatment and compliance coefficients, os_type_linux and revenue_pre are the only coefficients substantially different from 0 in both. However, only a very small proportion of users in the experiment are Linux users, and the distribution of revenue is very positively skewed. This justifies the minor difference between the DMLATEIV and DRIV estimates. Moreover, we fit a shallow, heavily regularized random forest and interpret it using Shapley Additive Explanations (SHAP) [24]. SHAP gave directionally similar impact of each feature on the effect (Figure 1). However, since we constrained the model to have depth at most one, it essentially gives the features in order of importance if we were to split the population based on a single feature. This justifies why the order of importance of features in the forest is not in the same order as the magnitude of the rank in the linear model, since they have different interpretations. The features picked up by the forest intuitively make sense since an already highly engaged member of TripAdvisor, or a user who has recently made a booking, is less likely to further increase their visits to TripAdvisor. Using gradient boosting nuisance models, we show that many inferences remain similar (Figure 3 in Appendix). The most notable changes in heterogeneity were for features which have a highly skewed distribution (e.g. visits to specific pages on TripAdvisor), or which appear rarely in the data (e.g. Linux users). The linear CATE projection model coefficients are largely similar for both residualization models (except the Linux operating system feature, which appears rarely in the data). Moving to a random forest for the CATE projection model with SHAP presents greater differences, especially for the highly skewed features.

**Similar instrument from a recent experiment** A recent 2019 A/B test of the same membership sign-up process provided another viable instrument. This 21-day A/B test included a much larger, more diverse population of users than in 2018 due to fewer restrictions for eligibility (see Sec. B.2 for

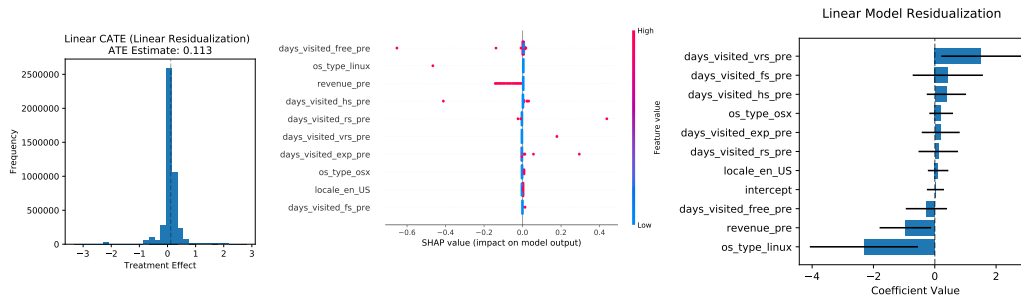

Figure 1: (From left to right) Linear CATE projection, SHAP summary of random forest CATE projection, Linear CATE projection coefficients. Using linear nuisance models.

details). We apply DRIV with gradient boosting residualization models and a linear projection of the CATE. The CATE distribution has generally higher values compared to the 2018 experiment which reflects the different experimental population. In particular, users in the 2018 experiment had much higher engagement and significantly higher revenue in the pre-experiment period. This was largely because users were only included in the 2018 experiment on their *second* visit. The higher baseline naturally makes it more difficult to achieve high treatment effects, explaining the generally lower CATE distribution in the 2018 experiment. We note that, unlike in 2018, the revenue coefficient is no longer significant. We again attribute this to the much higher revenue baseline in 2018. Despite the population differences, however, *we observe "days_visited_vrs_pre" continues to have a very significant positive association*. "days_visited_exp_pre" now also appears to have a significantly positive association, as does the iPhone device (which was not a feature in the 2018 experiment). The inclusion of iPhone users is another big domain shift in the two experiments.

**Policy recommendations for Trip Advisor** Our results offer several policy implications for Trip Advisor. Firstly, encourage iPhone users, and users who frequent vacation rentals pages to sign-up for membership. These users exhibited high treatment effects from membership. For frequent visitors to vacation rentals pages, this effect was robust across residualization models, CATE projections, and even different instruments (e.g. by providing stronger encouragements for sign-up on particular sub-pages). Secondly, find ways to improve the membership offering for users who are already engaged: e.g. recently made a booking (high revenue_pre), were already frequent visitors (high days_visited_free_pre).

**Validation on Semi-Synthetic Data** In Appendix C, we validate the correctness of ATE and CATE from DRIV, by creating a semi-synthetic dataset with the same variables and such that the marginal distribution of each variable looks similar to the TripAdvisor data, but where we know the true effect model. We find that DRIV recovers a good estimate of the ATE. The CATE of DRIV with linear regression as final stage also recovers the true coefficients, and a random forest final stage picks the correct factors of heterogeneity as most important features. Moreover, coverage of DRIV ATE confidence intervals is almost nominal at 94%, while DMLATEIV can be very biased and have 0 coverage. [6]

## 4 Estimating the Effect of Schooling on Wages

The causal impact of schooling on wages has been studied at length in Economics (see [14], [6], [7], [17]), and although it is generally agreed that there is a positive impact, it is difficult to obtain a consistent estimate of the effect due to self-selection into education levels. To account for this endogeneity, Card ([6]) proposes using proximity to a 4-year college as an IV for schooling. We analyze Card's data from the Nat. Long. Survey of Young Men (NLSYM, 1966) to estimate the ATE of education on wages and find sources of heterogeneity. We describe the NLSYM data in depth in Appendix D. At high level, the data contains 3,010 rows with 22 mostly binary covariates $X$, log wages ($y$), years of schooling ($T$), and 4-year college proximity indicator ($Z$). We apply DMLATEIV and DRIV with linear (LM) or gradient boosted (GBM) nuisance models to estimate the ATE (Table 2 and Table 8 in Appx. D). While the DMLATEIV results are consistent with Card's $(0.134, [0.026, 0.242]\ 95\%$ CI), this estimate is likely biased in the presence of compliance and effect heterogeneity (see Sec. 2). The DRIV ATE estimates, albeit lower, still lie within the 95% CI of the

|  |  | Observational Data | | Semi-Synthetic Data | | |
|---|---|---|---|---|---|---|
| Nuisance | Method | ATE Est | 95% CI | ATE Est | 95% CI | Cover ‡ |
| LM | DMLATEIV | 0.137 | [0.027, 0.248] | 0.654 | [0.621, 0.687] | 10% |
| LM | DRIV | 0.065 | [-0.02, 0.151] | 0.587 | [0.521, 0.652]† | 92% |

† Contains the true ATE (0.609)      ‡ Coverage for 95% CI over 100 Monte Carlo simulations

Table 2: NLSYM ATE Estimates for Observational and Semi-synthetic Data

DML ATE. We study effect heterogeneity with a shallow random forest an the last stage of DRIV. Fig. 2 depicts the spread of treatment effects, and the important features selected. Most effects (89%) are positive, with very few very negative outliers. The heterogeneity is driven mainly by parental education variables. We project the DRIV treatment effect on the mother's education variable to study this effect. In fig. 2, we note that treatment effects are highest among children of less educated mothers. This pattern has also been observed in [6] and [17].

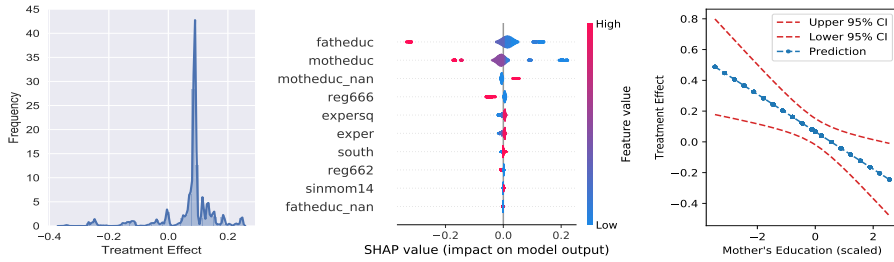

Figure 2: Treatment effect distribution, heterogeneity features, and linear projection on mother's education.

**Semi-synthetic Data Results.** We created semi-synthetic data from the NLSYM covariates $X$ and instrument $Z$, with generated treatments and outcomes based on known compliance and treatment functions (see Appx. D for details). In Table 2, we see that DMLATEIV ATE (true ATE=0.609) is upwards biased and has poor coverage over 100 runs, whereas the DRIV ATE is less biased and has overall good coverage. With DRIV, we also recover the correct $\theta(X)$ coefficients: $0.142$ ($[0.037, 0.245]$ 95% CI) vs 0.1, $0.049$ ($[0.015, 0.083]$) vs 0.05, and $-0.147$ ($[-0.365, 0.072]$) vs $-0.1$.

# 5 Acknowledgements

We thank Jeff Palmucci, Brett Malone, Baskar Mohan, Molly Steinkrauss, Gwyn Fisher and Matthew Dacey from TripAdvisor for their support and assistance in making this collaboration possible.

## Footnotes

[1]Prototype code for all the algorithms presented and the synthetic data experimental study can be found at https://github.com/Microsoft/EconML/tree/master/prototypes/dml_iv.

[2]Implicitly our moment condition abstracts away the low level conditions that allow one to interpret the parameter that satisfies the moment condition as causal. For instance, in the case of binary instruments and binary treatments to interpret the solution to the moment condition as the causal effect, one requires a monotonicity condition on the compliance structure [18], i.e. if a unit does not take the treatment when recommended, it would have also not taken the treatment without the recommendation. However, the fact that we also condition on $X$ and we estimate an effect conditional on $X$ only requires these conditions to hold conditional on $X$, i.e. the directionality of the effect of the instrument on the treatment can change for different $X$'s. So the requirements are milder. This weakening has also been observed in prior work in econometrics [19].

[3]For Double Machine Learning ATE estimation with Instrumental Variables.

[4] This corollary follows by small modifications of the proofs of Theorem 1 and Theorem 3 of [13] that accounts for the non-orthogonality w.r.t. $h$, so we omit its proof.

[5]We attempted to use the R implementation of Generalized Random Forests (GRF)[4] to compare with our results. However, we could not fit due to the size of the data and insufficient memory errors (with 64GB RAM).

[6]Results on the coverage experiment can be recovered by running `coverage.py` followed by post-processing with `post_processing.ipynb` at https://github.com/microsoft/EconML/tree/master/prototypes/dml_iv. Single synthetic instance results on the quality of the recovered estimates and comparisons with benchmark approaches can be found in `TA_DGP_Analysis.ipynb` at the same location.

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
