[Supplementary Material · heterogeneous_te_instruments_appendix.pdf]

## A Proof of Main Lemmas

Before we prove our two main lemmas we define the concept of an orthogonal loss. Consider a loss function $L(\theta; g)$ that depends on a target model $\theta \in \Theta$ and nuisance model $g \in G$.

**Definition 1** (Directional Derivative). *Let $V$ be a vector space of functions. For a functional $F : V \to \mathbb{R}$, we define the derivative operator*

$$D_g F(g)[\nu] = \frac{d}{dt} F(g + t\nu) \mid_{t=0},$$

*for a pair of functions $g, \nu \in V$. Likewise, we define*

$$D_g^k F(g)[\nu_1, \ldots, \nu_k] = \frac{\partial^k}{\partial t_1 \ldots \partial t_k} F(g + t_1 \nu_1 + \ldots + t_k \nu_k) \mid_{t_1 = \cdots = t_k = 0}.$$

When considering a functional in two arguments, e.g. $F(\theta, g)$, we will write $D_g F(\theta, g)$ and $D_\theta F(\theta, g)$ to make the argument with respect to which the derivative is taken explicit.

**Definition 2** (Orthogonal Loss). *The population risk $L(\theta; g)$ is* orthogonal, *if:*

$$D_g D_\theta L(\theta_0; g_0)[\theta - \theta_0, g - g_0] = 0 \quad \forall \theta \in \Theta, \forall g \in G. \tag{9}$$

**Definition 3** (Strong Convexity in Prediction). *The population risk $L(\theta; g)$ is strongly convex with respect to the prediction, if:*

$$D_\theta^2 L(\bar{\theta}, g)[\theta - \theta_0, \theta - \theta_0] \geq \lambda \|\theta - \theta_0\|_2^2 \quad \forall \theta \in \Theta, \ \forall g \in G, \ \forall \bar{\theta} \in star(\Theta, \theta_0).$$

*where:*

$$star(\Theta, \theta) = \{t \cdot \theta + (1 - t) \cdot \theta' : \theta' \in \Theta, t \in [0, 1]\}. \tag{10}$$

### A.1 Proof of Lemma 1

*Proof.* We show that the expected directional derivative of the moment (directional derivative of the loss with respect to $\theta(X)$) conditional on $X$, with respect to each of the nuisance functions is equal to zero, when evaluated at the true nuisance and target functions. The directional derivative of the loss with respect to direction $\nu = \theta' - \theta$ and evaluated at parameter $\theta$ is:

$$\mathbb{E}[m^1(X; \theta(X), q(X), p(X), h) \cdot \nu(X)]$$

where:

$$m^1(X; \theta(X), q(X), p(X), h) = -2 \mathbb{E}[(Y - q(X) - \theta(X) (h(Z, X) - p(X))) (h(Z, X) - p(X)) \mid X]$$

To show orthogonality with respect to $p, q$, it suffices to show that the classical derivative of $m^1$ with respect to the inputs $p(X)$ and $q(X)$ is zero, when evaluated at the true nuisance and target parameters:

$$\nabla_{q(X)} m^1(X; \theta_0(X), q_0(X), p_0(X), h_0) := -2 \mathbb{E}[h_0(Z, X) - p_0(X) \mid X] = 0$$

$$\begin{aligned}
\nabla_{p(X)} m^1(X; \theta_0(X), q_0(X), p_0(X), h_0) := & -2 \theta_0(X) \mathbb{E}[h_0(Z, X) - p_0(X) \mid X] \\
& + 2 \mathbb{E}[Y - q_0(X) - \theta_0(X) (h_0(Z, X) - p_0(X)) \mid X] \\
= & 0
\end{aligned}$$

Where in both equations we invoked the conditional moment restrictions to claim that they are equal to zero.

To prove orthogonality with respect to $h$ we need to show that the directional derivative of $m^1$ with respect to $h$ is zero. We cannot reduce it to a classical derivative condition, since $h$ takes as input the variable $Z$ which is not part of the conditioning set of the moment $m^1$. However, we see that this directional derivative evaluated at $h_0$ and at a direction $\nu = h - h_0$, is not zero:

$$\begin{aligned}
\mathcal{D}_h m^1(X; \theta_0, q_0, p_0, h_0)[\nu] := & 2 \theta_0(X) \mathbb{E}[(h_0(Z, X) - p_0(X)) \nu(Z, X) \mid X] \\
& + 2 \mathbb{E}[(Y - q_0(X) - \theta_0(X) (h_0(Z, X) - p_0(X))) \nu(Z, X) \mid X] \\
= & 2 \theta_0(X) \mathbb{E}[(h_0(Z, X) - p_0(X)) \nu(Z, X) \mid X]
\end{aligned}$$

The last quantity is not necessarily zero, since $\mathbb{E}[h_0(Z,X) - p_0(X) \mid Z, X] \neq 0$. This finding is reasonable since we are using $h(Z,X)$ as our regressor. Hence, any error in the measurement of the regressor should directly propagate to an error in $\theta(X)$. The quantity would have been zero if the residual error from the first stage function $h(Z,X) - h_0(Z,X)$ was independent of the residual randomness $h_0(Z,X) - p_0(X)$, conditional on $X$. However, the two in general can be correlated: the second quantity measures how far is $h_0(Z,X)$ from each mean $p_0(X) = \mathbb{E}[h_0(Z,X) \mid X]$, while the first quantity measures how far is the estimate $h(Z,X)$ from $h_0(Z,X)$. It is highly probable that when $Z$ takes values that lead to a large deviation from the mean treatment, then these are also the values of $Z$ for which the first stage model makes more mistakes. $\square$

## A.2 Proof of Lemma 3

*Proof.* We show that the expected derivative of the moment (derivative of the loss with respect to $\theta(X)$) conditional on $X$, with respect to each of the nuisance functions is equal to zero, when evaluated at the true nuisance and target functions. The directional derivative of the loss with respect to direction $\nu = \theta' - \theta$ and evaluated at parameter $\theta$ is:

$$-2\,\mathbb{E}[m^2(X; \theta(X), g(X)) \cdot \nu(X)]$$

where $g(X) = (\hat{\theta}(X), p(X), q(X), r(X), \beta(X))$ and :

$$m^2(X; \theta(X), g(X)) = \mathbb{E}\left[\hat{\theta}(X) + \frac{(Y - q(X) - \hat{\theta}(X)\,(T - p(X)))\,(Z - r(X))}{\beta(X)} - \theta(X) \,\Big|\, X\right]$$

To show orthogonality with respect to the nuisance functions $g$, it suffices to show that the classical derivative of $m^1$ with respect to each component of $g(X)$ is zero, when evaluated at the true nuisance and target parameters:

$$\nabla_{\hat{\theta}(X)} m^2(X; \theta_0(X), g_0(X)) := \mathbb{E}\left[1 - \frac{(T - p_0(X))\,(Z - r_0(X))}{\beta_0(X)} \mid X\right] = 0$$

$$= 1 - \frac{\mathbb{E}[(T - p_0(X))\,(Z - r_0(X))]}{\beta_0(X)} = 0$$

$$\nabla_{p(X)} m^1(X; \theta_0(X), g_0(X)) := \theta_0(X)\,\frac{\mathbb{E}[Z - p_0(X) \mid X]}{\beta_0(X)} = 0$$

$$\nabla_{q(X)} m^1(X; \theta_0(X), g_0(X)) := -\frac{\mathbb{E}[Z - r_0(X) \mid X]}{\beta_0(X)} = 0$$

$$\nabla_{r(X)} m^1(X; \theta_0(X), g_0(X)) := \theta_0(X)\,\frac{\mathbb{E}[T - p_0(X) \mid X]}{\beta_0(X)} = 0$$

$$\nabla_{\beta(X)} m^1(X; \theta_0(X), g_0(X)) := -\frac{\mathbb{E}\left[(Y - q_0(X) - \theta_0(X)\,(T - p_0(X)))\,(Z - r_0(X))\right]}{\beta_0(X)^2}$$

$$= -\frac{\mathbb{E}\left[\mathbb{E}[Y - q_0(X) - \theta_0(X)\,(T - p_0(X)) \mid Z, X]\,(Z - r_0(X))\right]}{\beta_0(X)^2}$$

$$= -\frac{\mathbb{E}\left[\mathbb{E}[Y - \theta_0(X)\,T - f_0(X) \mid Z, X]\,(Z - r_0(X))\right]}{\beta_0(X)^2} = 0$$

Where in all equations we invoked the conditional moment restrictions and the definitions of the true nuisance functions to claim that they are equal to zero.

Moreover, observe that the second directional derivative of the loss with respect to $\theta$ and for a direction $\nu = \theta' - \theta$ is equal to:

$$2\mathbb{E}[\nu(X)^T \nu(X)] \geq 2\|\nu\|_2^2 \tag{11}$$

Thus the loss is 2-strongly convex. $\square$

## B   W Data and Analysis

### B.1   Model Details and Parameters

**Residualization Models**

- LASSO regression and logistic regression with an L2 penalty using the Python sklearn library. For each cross-fitted fold, 3-Fold cross-validation was used to select the regularization parameter based on minimizing RMSE and log-loss.

- Gradient boosting (GB) regression and classification using the XGBoost library.[8] 100 estimators were used, with a minimum child weight of 20, and gamma set to 0.1. A 10% validation set was used for early stopping based on RMSE and log-loss.

The gradient boosting models from sklearn also yielded substantially similar results to XGBoost.

**Random Forest** We use a shallow, heavily regularized random forest for projection of the CATE. Parameters used: 1,000 trees, a minimum leaf size of 20,000, and a maximum depth size of 1. The heavy regularization is required in order to ensure stability of the CATE estimates.

**Linear Compliance Model** Using the 2018 experiment data and linear residualization models, the compliance quantity $E[T \cdot Z|X] - E[T|X] \cdot E[Z|X]$ (despite the logistic function) is well-approximated by a linear regression. We use this approximation for interpreting the coefficents of the fitted model (Figure 4).

## B.2 Additional Data Description and Preparation

Full description of the data in Table 3. The criteria for eligibility required that users were not existing members of W before the experimental period; visited W through a desktop browser during the experimental period; and visited W at least twice during the experimental period. The first visit did not activate the test functionality. Group assignment was determined randomly with equal probability, resulting in $n_A = 2,303,658$ in group A, and $n_B = 2,302,383$ in group B.

We transform the operating system categorical variable using one-hot encoding and drop the "Windows" level to use as the baseline. In addition, the co-variates are normalized uniformly over 1,000 quantiles, resulting in a $X_i \in [0,1]^{10}$ co-variate vector for each user to be used for both residualization and effect heterogeneity.

For confidentiality reasons, we report the ATE and CATE results normalized by $\hat{\mu}_B$, the mean number of days visited by users in group B of the A/B experiment. A treatment effect of 1 unit is therefore equal to $\hat{\mu}_B$ additional days visited.

| | |
|---|---|
| revenue_pre | Total revenue in dollars generated by the user in the pre-experimental period |
| days_visited_free_pre | Count of the days the user visited the W through free channels (e.g. email) in the pre-experimental period (0-28) |
| days_visited_hs_pre | Count of the days the user visited the hotels pages of W in the pre-experimental period (0-28) |
| days_visited_exp_pre | Count of the days the user visited the experiences pages of W in the pre-experimental period (0-28) |
| days_visited_rs_pre | Count of the days the user visited the restaurants pages of W in the pre-experimental period (0-28) |
| days_visited_vrs_pre | Count of the days the user visited the vacation rentals pages of W in the pre-experimental period (0-28) |
| days_visited_fs_pre | Count of the days the user visited the flights pages of W in the pre-experimental period (0-28) |
| os_type | Categorical variable for the user's operating system (3 levels) |
| locale_en_US | Binary variable indicating whether the user was from the en_US locale |
| Y | Outcome measurement, count of the number of total days the user visited W |
| T | Treatment, binary variable of whether the user became a member during the experimental period |
| Z | Instrument, binary variable of the user's group assignment in the A/B test |

Table 3: Definition of variables in the 2018 experimental data from W

**Additional details about the 2019 experiment** There were some key differences compared to the 2018 A/B test:

- the test was run for 3 weeks instead of 2;
- the test functionality was displayed on both desktop and mobile platforms across nearly all pages of W (i.e. not just the homepage);
- first-time visitors were eligible for the test; and
- the sample size was much larger at $n = 84,657,263$ users. We use a sample of $n_S = 10,158,871$ users stratified by A/B test group allocation for computational reasons.

## B.3 Additional Figures of Experimental Results

Figure 3: (From left to right) Linear CATE projection, SHAP summary of random forest CATE projection, Linear CATE projection coefficients. Using gradient boosting nuisance models.

Figure 4: Coefficients of the linear model approximation of the compliance quantity $E[T \cdot Z|X] - E[T|X] \cdot E[Z|X]$. Using linear models for nuisance.

Figure 5: Coefficients of the CATE linear projection model using DRIV with gradient boosting residualization on the W 2019 experiment data.

Figure 6: Coefficients of the linear CATE projection model for DRIV

(a) Linear nuisance          (b) GB nuisance

Figure 7: SHAP summary plot of the DRIV random forest CATE projection model

Figure 8: Distribution of CATE estimates using linear nuisance models.

Figure 9: Distribution of CATE estimates using gradient boosting nuisance models.

## C   Semi-Synthetic Data Analysis for W Data

**W Semi-synthetic Data Results.** In order to validate the correctness of ATE and CATE from DRIV model, we consider a semi-synthetic data generating process that looks similar in structure to W

data. The covariates have the same schema but are generated from fixed marginal distributions. The instrument corresponds to a fully randomized recommendation of treatment. And the compliance rates are generated to be similar with the experiment. This probability depends both on the observed feature $X$ and an unobserved confounder that has a direct effect on the outcome. The X covariates and DGP are given by:

| Covariate | Distribution |
|---|---|
| days_visited_free_pre, days_visited_hs_pre, days_visited_rs_pre, days_visited_exp_pre, days_visited_vrs_pre, days_visited_fs_pre | $X \sim \text{U}\{0, 28\}$ |
| locale_US | $X \sim \text{Bernoulli}(p = .5)$ |
| os_type | $X \sim \{OSX, Windows, Linux\}$ |
| revenue_pre | $X \sim \text{Lognormal}(\mu = 0, \sigma = 3)$ |

Table 4: Data Generation of Covariates X

$$
\begin{aligned}
Z &\sim \text{Bernoulli}(p = .5) && \text{(Instrument)} \\
\nu &\sim \text{U}[0, 10] && \text{(Unobserved confounder)} \\
C &\sim \text{Bernoulli}(p = 0.017 \cdot \text{Logistic}(0.1 \cdot (X[0] + \nu))) && \text{(Compliers when recommended)} \\
C0 &\sim \text{Bernoulli}(p = 0.006) && \text{(Non-Compliers when not recommended)} \\
T &\sim C \cdot Z + C0 \cdot (1 - Z) && \text{(Treatment)} \\
y &\sim \theta(X) \cdot (T + 0.1 \cdot \nu) + 0.4 \cdot X[0] + 2 \cdot \text{U}[0, 1] && \text{(Outcome)}
\end{aligned}
$$

Moreover, the treatment effect function is predefined here, which depends on the feature "days_visited_free_pre"(X[0]) and "locale_US"(X[6])

$$
\theta(X) = 0.2 + 0.1 \cdot X[0] - 2.7 \cdot X[6] \tag{CATE}
$$

We rerun the same experiments with 4 million samples. In table 5, it shows that both DMLATEIV and DRIV with either linear or GBM nuisance estimators, their ATE CI can recover the true estimate of ATE. Moreover, we validate the CATE via DRIV. In figure 10 and 11, we can see that DRIV with linear regression as final stage recovers the true coefficient from CATE, and the last stage model using random forest also picks the correct factor of heterogeneity as the most important features.

| Nuisance Models | Method | True ATE | ATE Estimate | 95% CI |
|---|---|---|---|---|
| Linear Models | DMLATEIV | 0.249 | 0.336 | [0.186, 0.487] |
| Linear Models | DRIV with constant | 0.249 | 0.166 | [-0.025, 0.358] |
| Gradient Boosting Models | DMLATEIV | 0.249 | 0.342 | [0.191, 0.492 ] |
| Gradient Boosting Models | DRIV with constant | 0.249 | 0.136 | [-0.060, 0.332 ] |

Table 5: ATE Estimates for Semi-Synthetic Data (n=4,000,000,coef=0.1)

Figure 10: (from left to right) CATE projection on X[0] and X[6] by linear final stage model, CATE projection on X[0] and X[6] by RF final model, SHAP summary of RF CATE projection. (n=4,000,000, coef=0.1, linear nuisance models)

Figure 11: Same plot for n=4,000,000, coef=0.1, GBM nuisance models

We also run some other experiments with different sample size $n$ and different level of endogeneity (the coefficient of variable $\nu$) to learn the consistency of ATE for these two models. we can see from the table and figures below that all of their CI covers the true estimate of ATE, but with the increase of $n$ and the decrease of the endogeneity coefficient, the ATE of DMLATEIV is more biased.

| Nuisance Models | Method | True ATE | ATE Estimate | 95% CI |
|---|---|---|---|---|
| Linear Models | DMLATEIV | 0.249 | 0.349 | [0.230, 0.468] |
| Linear Models | DRIV with constant | 0.249 | 0.197 | [0.044, 0.350] |
| Gradient Boosting Models | DMLATEIV | 0.249 | 0.354 | [0.235, 0.473] |
| Gradient Boosting Models | DRIV with constant | 0.249 | 0.179 | [0.023, 0.335] |

Table 6: ATE Estimates for Semi-Synthetic Data (n=4,000,000,coef=0.01)

Figure 12: Same plot for n=4,000,000, coef=0.01, Linear nuisance models

Figure 13: Same plot for n=4,000,000, coef=0.01, GBM nuisance models

| Nuisance Models | Method | True ATE | ATE Estimate | 95% CI |
|---|---|---|---|---|
| Linear Models | DMLATEIV | 0.250 | 0.350 | [0.045, 0.655] |
| Linear Models | DRIV with constant | 0.250 | 0.167 | [-0.222, 0.556] |
| Gradient Boosting Models | DMLATEIV | 0.250 | 0.344 | [0.040, 0.648] |
| Gradient Boosting Models | DRIV with constant | 0.250 | 0.253 | [-0.212, 0.718] |

Table 7: ATE Estimates for Semi-Synthetic Data (n=1,000,000,coef=0.1)

Figure 14: Same plot for n=1,000,000, coef=0.1, Linear nuisance models

Figure 15: Same plot for n=1,000,000, coef=0.1, GBM nuisance models

**Coverage Experiment.** To further validate the consistency of DMLATEIV and DRIV under effect and compliance heterogeneity, we create a slightly different semi-synthetic dataset with stronger instrument and less samples ($n = 100,000$) to run 100 times Monte Carlo Simulations. The DGP is given by:

$$Z \sim \text{Bernoulli}(p = .5) \qquad \text{(Instrument)}$$
$$\nu \sim \text{U}[0, 10] \qquad \text{(Unobserved confounder)}$$
$$C \sim \text{Bernoulli}(p = 0.2 \cdot \text{Logistic}(0.1 \cdot (X[0] + \nu))) \qquad \text{(Compliers when recommended)}$$
$$C0 \sim \text{Bernoulli}(p = 0.1) \qquad \text{(Non-Compliers when not recommended)}$$
$$T \sim C \cdot Z + C0 \cdot (1 - Z) \qquad \text{(Treatment)}$$
$$y \sim \theta(X) \cdot (T + 0.2 \cdot \nu) + 0.1 \cdot X[0] + 0.1 \cdot \text{U}[0, 1] \qquad \text{(Outcome)}$$

Moreover,

$$\theta(X) = 0.8 + 0.5 \cdot X[0] - 3 \cdot X[7] \qquad \text{(CATE)}$$

It turns out that distribution of DMLATEIV ATE has smaller variance but larger bias with 0 coverage to the true ATE, while DRIV ATE are more converged to the true ATE with 94% coverage.

Figure 16: DMLATEIV VS. DRIV ATE Estimates across 100 Monte Carlo Experiments: (left) distribution of ATEs across experiments, (middle) qq-plot of distribution of DRIV ATE vs normal centered at true estimate, scaled by std of DRIV, (right) qq-plot of distribution of DMLATEIV ATE vs normal centered at true estimate, scaled by std of DMLATEIV.

## D  NLSYM Data Analysis

**NLSYM Data Results.** The NLSYM data is comprised of 3,010 entries from men ages 14-24 that were interviewed in 1966 and again in 1976. We use the covariates $X$ selected by Card: mother and father education, family composition at 14, workforce experience, indicators for black, region, southern residence and residence in an SMSA in 1966 and 1976. The outcome of interest $y$ is log wages, the treatment $T$ is the years of schooling, and the instrument $Z$ is an indicator of whether the participant grew up near a 4-year college.

|          |          | Observational Data | | Semi-Synthetic Data | | |
|----------|----------|---------|------------------|---------|-------------------|----------|
| Nuisance | Method   | ATE Est | 95% CI           | ATE Est | 95% CI            | Cover ‡  |
| LM       | DMLATEIV | 0.137   | [0.027, 0.248]   | 0.654   | [0.621, 0.687]    | 10%      |
| LM       | DRIV     | 0.065   | [-0.02, 0.151]   | 0.587   | [0.521, 0.652]†   | 92%      |
| GBM      | DMLATEIV | 0.138   | [0.025, 0.251]   | 0.645   | [0.613, 0.695]    | 30%      |
| GBM      | DRIV     | 0.052   | [-0.032, 0.136]  | 0.612   | [0.548, 0.677]†   | 86%      |

† Contains the true ATE (0.609)      ‡ Coverage for 95% CI over 100 Monte Carlo simulations

Table 8: NLSYM ATE Estimates for Observational and Semi-synthetic Data

**Semi-synthetic Data Results.** The NLSYM data is a relatively small dataset and $Z$ could potentially be a weak instrument, which could explain the large confidence intervals in the prior analysis. To disentangle these effects, we create semi-synthetic data from the NLSYM covariates $X$ and instrument $Z$, with generated treatments and outcomes based on known compliance and treatment functions. The data generating process for the semi-synthetic data is given by:

$$\nu \sim \text{U}[0, 1] \qquad \text{(Unobserved Confounder)}$$
$$C = c_0 \cdot X[4], \ c_0 \ (const) \sim \text{U}[0.2, 0.3] \qquad \text{(Compliance Level)}$$
$$T = C \cdot Z + g(X) + \nu \qquad \text{(Treatment)}$$
$$y \sim \theta(X) \cdot (T + \nu) + f(X) + \mathcal{N}(0, 0.1) \qquad \text{(Outcome)}$$

We create a realistic heterogeneous treatment effect that depends on the mother's education (X[4]) and whether the child was in the care of a single mother at age 14 (X[7], 10% of subjects):

$$\theta(X) = 0.1 + 0.05 \cdot X[4] - 0.1 \cdot X[7] \qquad \text{(CATE)}$$
$$f(X) = 0.05 \cdot X[4], \; g(X) = \; X[4] \qquad \text{(Nuissance Functions)}$$