[Reviews · NeurIPS 2019]

Reviewer 1



I believe this paper should be accepted to NeurIPS. Originality: This paper combines a number of known ideas into a very nice framework to do heterogeneous treatment effect estimation in the IV setting. Quality: The paper is technically correct. The experiments are well motivated and well done. Clarity: The paper is very clear and easy to understand (my one comment is that I had to rewrite the equations for myself ignoring the intercept terms to really understand what was going on). Significance: This is a method with lots of potential applications in both academic and industrial practice.

Reviewer 2



The authors develop new algorithms for instrumental variables based on the orthogonal ML technique. The paper considers the IV problem in terms of moment conditions and derive conditions where the target quantity of interest has a good rate of estimation. Then the evaluation is done on a number of experimental settings on semi-synthetic and real data. The theoretical contributions could be better explained instead of purely linking it away to existing literature; please consider adding full consistency proof for completeness. It would also serve the reader if a larger discussion the original double ML work and neyman-orthogonality was included. The experimental results look promising but I think it is premature to judge them without too many benchmarks and baselines. could the authors explain the width of large width of the confidence intervals? Is it a data artefact or something practitioners should keep in mind? My main issue is that interaction between a few elements of the theorem like the functions g, delta haven't been explored in the semi-synthetic experiments. To add to the validity of the proposed methods, more functionals relationships between the covariates and the outcome should be explored and the improvement of the method over other flexible IV methods should be demonstrated. Overall this work I think addresses an important gap in the literature of observational causal estimation where IVs are leveraged. ---- POST REBUTTAL ---- The authors addressed the issues raised. I now have more faith in the presented experimental results and the discussion around it. As the authors point out, weak IVs do present significant problems. They will add more experiments to provide a full context for DMLIV's performance. Please consider adding a quick discussion about the orthogonality conditions to the appendix. Maybe discuss how they allow practitioners to trade-off interpretability and performance by specifying certain parts of the model; like the partially linear model, for example. I've raised the score accordingly.

Reviewer 3



The paper is clear and seems correct as far as I can tell, but I found it may not be very accessible to those who are not familiar with the double machine learning approach. The paper proposes an extension of the two-stage least squares method (DMLATEIV) that allows arbitrary models. The extension itself seems quite straightforward, and the significance of this contribution is limited. The authors show that DMLATEIV does not satisfy the Neyman orthogonality, meaning that it is sensitive to the errors in the first-stage estimation. To mitigate the weakness, the authors suggest modifying the estimator by the doubly-robust approach and show that the loss for the modification is Neyman orthogonal, and the resulting estimator is robust to the estimation errors of the nuisance estimators. This seems to be a very important and useful result. However, the application of the doubly robust approach to instrumental variable regression is not totally new, and the resulting estimator is known to be Neyman orthogonal according to [V. Chernozhukov, D. Chetverikov, et al. 2017. Double/Debiased/Neyman Machine Learning of Treatment Effects. arXiv 1701.08687]: "Neyman-orthogonal scores are readily available for both the ATE and ATTE – one can employ the doubly robust/efficient scores of Robins and Rotnitzky (1995) and Hahn (1998), which are automatically Neyman orthogonal." Nevertheless, I could not find any previous work on this topic for the heterogeneous setting. The proposed method is applied to a real-world treatment effect problem and other datasets. The results look great, but there is no comparison with other methods, which is a weak point of the paper. ===== Update after the authors' response: The authors' response has addressed my major concern with the novelty and the significance of the technical contributions. As the authors claim, the paper in fact presents a solution to an open question raised in [Xinkun Nie and Stefan Wager, 2017. Quasi-Oracle Estimation of Heterogeneous Treatment Effects.]. The authors are encouraged to report the results for the experimental comparison with DeepIV mentioned in the authors' feedback.

[Author Response · NeurIPS 2019]

We thank the reviewers for their thoughtful comments and suggestions and we respond below to some concrete questions/comments that were raised.

**Response to Reviewer #1.**

*Sample splitting.* We agree that it was our omission not to point out more clearly the relevant references on sample splitting. The most related is the sample splitting performed in prior papers that use the notion of Neyman orthogonality such as [1, 2]. We will add the relevant discussion and the relation to the papers recommended by the reviewer in the revision.

**Response to Reviewer #2.**

*Width of confidence intervals.* The intervals are large for two primary reasons: 1) the strength of the instrument, which is a typical source of high variance in IV regression, 2) the correlation among the features. Despite these sources, the number of samples was still large enough to identify several statistically significant non-zero coefficients.

*Further semi-synthetic experiments.* We have conducted several experiments with several functional forms for the HTE. We chose to depict a representative subset in the main text and the supplementary material, focusing primarily on the real world data. We will definitely augment with more semi-synthetic analysis in the revision and we have already included in our submission and will make public our code, that contains an easy to use Jupyter notebook, where one can play with the functional form of the HTE (see W_DGP_analysis.ipynb). For instance, we depict below one example of a DGP with a step-wise and discontinuous HTE and how our DRIV with a random forest final stage HTE model performs as a quick non-linear qualitative example (there are 9 features of which only 2 are relevant. One is binary and one continuous; the two dotted lines correspond to the HTE functions for the two values of the binary feature and the shaded lines correspond to the estimates recovered by DRIV).

*Further benchmarks.* We compared primarily to the orthogonal IV approach for ATE estimation of [2]. This is denoted as DMLATEIV. We found DeepIV to be unstable for our problem and we did not include the results. One major discrepancy is that the existing implementation of DeepIV uses mixture density networks to fit the distribution of the treatment conditional on the instrument. However this is an overkill when the treatment is binary (which is our primary case). In that case the advantage of DeepIV (which is primarily that it can capture non-linear relationships with respect to the treatment), is lost, since the outcome is linear in the treatment, without loss of generality (due to the binary nature). In such settings it suffices to just fit a regression that predicts the mean treatment conditional on the instrument and the features, as opposed to a distribution. This is essentially our DMLIV method; on top of that DMLIV also performs residualization which leads to extra robustness. For these reasons we omitted DeepIV experiments. However, we will definitely add such experiments in the revision.

*Title.* We will change to a more elaborate title, e.g. "Orthogonal Machine Learning Estimation of Heterogeneous Treatment Effects with Instruments".

**Response to Reviewer #3.** We note that the main methodological novelty of the the DRIV method is to produce a Neyman orthogonal loss for the HTE estimation problem with instruments. The existence of such a loss was not known in the econometrics literature (for instance it was explicitly posed as an open question in [3]). The comment of [2] on the availability of Neyman orthogonal scores for ATE via the work of Robins and Rotnitzky is primarily for the conditional exogeneity setup and not for the IV setup with an unobserved confounder. Doubly robust estimates for the IV setup were developed in the work of [4]. However, this work assumes constant effect. We will improve upon our discussion on related work in our introduction and how our method contributes to the econometrics literature.

# References

[1] Victor Chernozhukov, Juan Carlos Escanciano, Hidehiko Ichimura, Whitney K. Newey, and James M. Robins. Locally Robust Semiparametric Estimation. *arXiv e-prints*, page arXiv:1608.00033, July 2016.

[2] Victor Chernozhukov, Denis Chetverikov, Mert Demirer, Esther Duflo, Christian Hansen, Whitney Newey, and James Robins. Double/debiased machine learning for treatment and structural parameters. *The Econometrics Journal*, 21(1):C1–C68, 2018.

[3] Xinkun Nie and Stefan Wager. Quasi-oracle estimation of heterogeneous treatment effects. *arXiv preprint arXiv:1712.04912*, 2017.

[4] Ryo Okui, Dylan S Small, Zhiqiang Tan, and James M Robins. Doubly robust instrumental variable regression. *Statistica Sinica*, pages 173–205, 2012.


[Meta-Review · NeurIPS 2019]

This paper solves an open problem, giving for the first time a provable method for efficiently using arbitrary ML algorithms for estimating heterogeneous effects in the instrumental variable scenario. Substantial experiments on several large datasets very nicely tie together theory to practice, in important and meaningful application areas. The reviewers have several proposal for making the paper clearer, which I trust the authors will follow. An important issue that must be clarified is the role of the monotonicity assumption given informally in the second paragraph of the paper.